# Design, Construction, and Validation of an Experimental Electric Vehicle with Trajectory Tracking

**DOI:** 10.3390/s24092769

**Published:** 2024-04-26

**Authors:** Joel Artemio Morales Viscaya, Alejandro Israel Barranco Gutiérrez, Gilberto González Gómez

**Affiliations:** Departamento de Estudios de Posgrado e Investigación, Tecnológico Nacional de México en Celaya (TecNM), Antonio García Cubas #600, Celaya 38010, Guanajuato, México; d2003026@itcelaya.edu.mx (J.A.M.V.); israel.barranco@itcelaya.edu.mx (A.I.B.G.)

**Keywords:** trajectory tracking, electric vehicle, control, embedded system, sensors

## Abstract

This research presents an experimental electric vehicle developed at the Tecnológico Nacional de México Celaya campus. It was decided to use a golf cart-type gasoline vehicle as a starting point. Initially, the body was removed, and the vehicle was electrified, meaning its engine was replaced with an electric one. Subsequently, sensors used to measure the vehicle states were placed, calibrated, and instrumented. Additionally, a mathematical model was developed along with a strategy for the parametric identification of this model. A communication scheme was implemented consisting of four slave devices responsible for controlling the accelerator, brake, steering wheel, and measuring the sensors related to odometry. The master device is responsible for communicating with the slaves, displaying information on a screen, creating a log, and implementing trajectory tracking techniques based on classical, geometric, and predictive control. Finally, the performance of the control algorithms implemented on the experimental prototype was compared in terms of tracking error and control input across three different types of trajectories: lane change, right-angle curve, and U-turn.

## 1. Introduction

An autonomous vehicle, commonly referred to as a self-driving car, is an automobile equipped with advanced technologies that enable it to perform certain operational and navigational functions without direct human intervention. These vehicles utilize a combination of sensors, cameras, radar systems, and sophisticated algorithms to accurately determine their current position and status, as well as to interpret the surrounding environment in real time. The primary objective of autonomous vehicles is to achieve a high level of automation, thus encompassing basic functionalities like cruise control to more advanced capabilities where the vehicle can autonomously handle complex driving tasks [1].

In the realm of modern transportation, autonomous vehicles stand as a groundbreaking advancement with the potential to revolutionize road safety. The advent of autonomous driving technology introduces a paradigm shift by significantly reducing the need for direct human intervention, which is often a contributing factor to traffic accidents. With the incorporation of advanced sensors, real-time data processing, and machine learning algorithms, autonomous vehicles possess the capability to perceive and respond to their surroundings with unparalleled precision. By mitigating human errors such as distraction, fatigue, or impaired driving, autonomous vehicles aim to create a safer driving environment. This is particularly significant in the context of Mexico, where road safety remains a pressing concern. The integration of these technologies holds the promise of a future where traffic accidents become increasingly rare, thus paving the way for a transportation landscape characterized by enhanced safety and efficiency. According to a study by IDTechExe [2], the market value of driverless vehicles will continue to grow at least until 2034. Furthermore, recent analyses by the information technology company Gartner suggest that autonomous vehicles are poised to increase their market share in the automotive industry within the next few years [3].

The motivation for this work stems from the absence of autonomous vehicle prototypes within educational institutions across Mexico, despite the current and anticipated growth of autonomous vehicles globally. With Mexico’s prominent position as a key player in the North American automotive industry [4], there is a pressing need to address this gap. Autonomous vehicle prototypes are crucial for facilitating testing and experimentation in fields related to trajectory tracking and autonomous vehicle technologies. By addressing this absence and developing a prototype, this work aims to fill the void in practical learning opportunities for students pursuing careers and subjects related to autonomous vehicles. Thus, the primary motivation of this work is to contribute to the advancement of autonomous vehicle research and education within Mexican educational institutions.

This research project centers on the construction, development, and validation of an experimental electric vehicle equipped with a trajectory tracking control system. The project entails several key steps, including the electrification of a golf cart-type vehicle; the precise placement, calibration, and instrumentation of sensors to measure the vehicle’s states; the parametric modeling and identification of the vehicle; and finally, the implementation, validation, and comparison of various trajectory tracking techniques. These techniques encompass classical, geometric, and predictive control methodologies, which are aimed at achieving optimal performance and accuracy in trajectory tracking.

The developed prototype offers several noteworthy advantages, particularly its cost-effectiveness. Notably, the trajectory tracking system is designed without relying on expensive or specialized sensors, thus contributing to its affordability. Furthermore, the communication scheme operates internally, thereby eliminating the need for dedicated wireless computing equipment; instead, communication is facilitated through Arduino and Raspberry Pi devices. Beyond its technological contributions to Mexico through construction and validation, the research also proposes a mathematical model for simulation and conducts an experimental comparison of various trajectory tracking techniques.

This document is structured as follows: Section 2 and Section 3 provide an overview of the prototype vehicle, thereby detailing the hardware, sensors, and communication scheme utilized. Section 4 briefly discusses the model and parametric identification techniques employed. Section 5 and Section 6 delve into the trajectory tracking algorithms applied and present the experimental results. Finally, Section 7 offers conclusions drawn from the findings and outlines potential avenues for future research.

## 2. Materials and Methods

### 2.1. The Prototype Electric Vehicle

The vehicle chosen as the prototype for this research project is based on the chassis of a gasoline-powered golf cart originally acquired by the Autotronics laboratory of the Tecnológico Nacional de México campus in Celaya. The primary objective was to electrify the vehicle, thus essentially converting or adapting it to operate using an electric motor in lieu of an internal combustion engine. Figure 1 showcases the vehicle chassis.

The original gasoline combustion engine was replaced by a series wound DC motor ES-15A-6, 48-72V D&D; an image of said motor can be seen in Figure 2. The characteristics of the motor used are the following:

D&D brand series winding-type motor; this DC motor can operate at voltages from 48 to 72 V. With the capacity to develop up to 25 horsepower (HP). Its characteristics include an operating voltage of 48–72 V, a rated capacity of 9 HP 72 V continuous, a maximum peak capacity of 25 HP, a main shaft of 7/8 in diameter, a motor diameter of 6.7 in, a motor length of 12.28 in, and a weight of 30 Kg approx.

To power the motor, six TROJAN T1275 150 Ah 12 V batteries were used, which were placed just below the seats on the chassis, as can be seen in Figure 3. The TROJAN T1275 150 Ah 12 V Battery is a 12 V voltage, monobloc-type battery that is composed of six 2 V cells delivering a total voltage of 12 V. It is a battery that requires special attention, as it is open lead acid. Every certain time, depending on its use, it is necessary to check its electrolyte levels.

An ALLTRAX controller model SR-72500 Series, 12–72 V, 500 A, was used as a voltage converter, whose specifications are the following:

Operating voltage from 12 to 72 VDC; programmable via USB port, 380 Amp continuous; 500 Amp peak; includes high-reliability fan. It has input for inductive type accelerator; resistive 0–5 KΩ; (± 10%) two or three wires. Resistive 5K–0 Ω (±10%), 0–5 V, 6–10.5 V. An image of the controller can be seen in Figure 4.

In the presented configuration, it is observed that the motor operates through a battery pack comprising six 12-volt units. A protective fuse and a contactor are integrated in series with the power supply to safeguard the controller against potential ignition sparks. Furthermore, an enable switch and a potentiometer, serving as the speed control mechanism for the engine—referred to colloquially as the “accelerator pedal”—is linked to the ALLTRAX controller. Notably, the motor’s connection to the battery’s negative terminal is facilitated via the controller, whereas the positive terminal is under the sole regulation of the contactor. The connection diagram of the motor and the voltage controller can be seen in Figure 5.

The doors and a fiberglass body were designed and added to the vehicle in such a way that the final prototype can be seen in Figure 6.

### 2.2. Hardware and Sensors

To carry out the instrumentation and control process of the experimental vehicle, it was decided to use the following hardware components:2 a3144 Hall effect sensors;2 MPLD2C10K linear slide potentiometers;1 KY-040 rotary encoder;1 HMC5883L magnetometer;1 MPU6050 accelerometer;1 Raspberry Pi 3;4 Arduino Unos;3 IBT-2 H bridges;1 16x2 LCD screen with i2C interface;3 electrical switches.

Two a3144 Hall effect sensors were placed, one on each of the rear tires, to measure their angular displacement and thereby estimate the vehicle’s speed.

In order to measure the angular displacement of the tires, 16 equally spaced neodymium magnets were placed on the tire rims. When a magnet passes by the Hall effect sensor, it detects the variation in the magnetic field.

The linear sliding potentiometers were positioned to measure the position of each of the vehicle’s pedals. When pressed, a metal piece is used to move the potentiometer’s axis alongside the pedal, as depicted in Figure 7, thus varying the electrical resistance.

Using an L-type metal support, a plastic wheel with a KY-040 rotary encoder attached to its rotation axis was affixed to the steering wheel’s rotation axis. This configuration allowed for the indirect measurement of the steering wheel’s angular displacement, as depicted in Figure 8. By measuring this displacement, the orientation of the front tires could be determined.

Both the HMC5883L magnetometer and the MPU6050 accelerometer were mounted outside the vehicle atop its structure. This positioning was chosen to minimize the influence of the electric motor on the orientation and acceleration measurements of the vehicle body. To provide some protection, both devices were housed beneath a small plastic cover that extends from the vehicle, as illustrated in Figure 9.

The HMC5883L magnetometer requires calibration to ensure accurate and reliable measurement of magnetic fields. In addition to positioning it outside the vehicle, it was necessary to determine the offsets for each magnetometer. These offsets represent the magnetic field values measured even when the sensor is not exposed to external magnetic fields. These offsets were subtracted from the magnetic field calculations in the program used for measurement. To accomplish this, an algorithm based on code available in helscream’s GitHub repository [5] on the Arduino IDE was utilized.

Furthermore, an enhanced offset adjustment was implemented using data collected from four different vehicle positions. Each position was obtained by rotating the vehicle by 90 degrees until completing a full 360-degree turn. To refine the offset, this adjustment was formulated as an optimization problem. The cost function for this optimization was defined as the difference between 90 degrees and the change in orientation calculated using the offset.

After determining the optimal offsets, achieved through a MATLAB script utilizing the fminsearch function, these offsets were subtracted from the measurements obtained by the Arduino Uno responsible for communicating with the sensor.

Furthermore, calibration of the MPU6050 gyro was essential to eliminate zero error. This error occurs when the sensor detects a small angle, even though it should register as completely level. To rectify this, an offset was applied to the raw accelerometer and gyroscope sensor readings. The adjustment was made until the gyroscope readings indicated zero rotation, and the accelerometer registered the acceleration due to gravity pointing directly downward.

An algorithm was used based on the code available in the GitHub i2cdevlib repository by Jeff Rowes [6] on the Arduino IDE that, to start the calibration, requires placing the MPU6050 module in a flat and level position and then sending any character on the serial monitor.

The 3 IBT-2 H bridges served to control the vehicle’s motors using the PWM (Pulse Width Modulation) technique. Additionally, the 16 × 2 LCD screen with i2C interface was employed to present real-time vehicle performance metrics, including speed, displacement, and other pertinent data during test runs.

The Raspberry Pi 3 and Arduino Uno devices were utilized for data acquisition from the aforementioned sensors and for sending control signals to the vehicle actuators. The details of this process will be discussed in the following section. The three electric switches served to facilitate communication with the Raspberry Pi 3 main board during the execution of the tracking algorithms.

## 3. Communication Scheme

To execute tracking control on the vehicle, two stages are essential:A preliminary stage involves an algorithm determining the desired values for both the pedals and the steering wheel to follow a specific trajectory.Subsequently, an internal stage consists of three algorithms responsible for adjusting the accelerator, brake, or steering wheel to the desired positions identified in the preceding stage.

To address this, and considering the processing limitations of the low-cost electronic devices employed, a master–slave scheme was adopted utilizing multiple devices.

The devised scheme entails four slave Arduino Uno devices communicating with a Raspberry Pi 3 master device. Each device is tasked with communicating with different sensors and executing various functions.

The master program operates on a Raspberry Pi 3 and is responsible for executing four functions concurrently using threads:Acquire information from each Arduino Uno device.Log sensor data into a CSV file during execution.Display pertinent information on the LCD screen, such as vehicle displacement and speed or details relevant to the conducted test.Employ control algorithms to determine desired values for the throttle, brake, and steering wheel, and transmit them to the respective Arduino Uno devices.

The general communication diagram can be seen in Figure 10.

### 3.1. Throttle Control and Brake Control

These devices are analogous; they fulfill identical functions, with one connected to the accelerator and the other to the brake. Their primary responsibilities include measuring the potentiometers, implementing a Kalman filter for position (to eliminate noise), and scaling the position values between 0 and 100 for standardization reasons.

In each iteration, the program first checks the serial buffer for numerical information ranging from zero to one hundred. If such information is present, it is stored as the desired position.

Subsequently, the error, its derivative, and integral are computed, followed by PID control using an IBT-2 H bridge to modulate the PWM signal, thus aligning the position with the desired one.

For synchronization with the master device, the <TimerOne.h> library was utilized to create a Timer1 object. The attachInterrupt function was then employed to assign an interrupt per clock, thereby ensuring that position values of the motors are sent via the serial port to the master program every hundred milliseconds.

### 3.2. Steering Control

This device, unlike the previous ones, not only communicates the position of the steering wheel and adjusts it to the desired position but also measures and transmits the readings of the previously calibrated HMC5883L magnetometer via the serial port.

The main loop begins by checking if a value exists in the serial communication. If a value within the range of integers between −38 and 38 is detected, it is interpreted as the desired direction of the steering wheel. It was previously determined that the resolution of the rotational encoder gives 38 positions from the center to the maximum position of the steering wheel.

Subsequently, an algorithm was created to control the position of the steering wheel with the following steps:If the error is of magnitude equal to or less than one, there is no control action (to avoid oscillatory vibration behavior between ±1 of the desired value, a tolerance of one position is given).If the error has remained constant, the control signal is gradually increased.If the error has had an oscillatory behavior, the control signal is gradually decreased.

Originally, a PID controller was considered; however, it was not possible to perform tuning that would produce desirable behavior. When the desired position was far from the current one, it produced, with high values for $K_{p}$, unwanted violent movements in the steering wheel, or, in the case of lower values in $K_{p}$, the torque was not enough to move the steering wheel.

Due to the presence of some mechanical imperfections in the steering wheel, the same torque that moved the steering wheel abruptly in some configurations could not do so in others, so it was decided to treat as particular cases if the pair did not move the steering wheel or if the movement was very abrupt and produced oscillatory behavior.

### 3.3. Odometry

The remaining Arduino Uno was used for measurements related to the calculation of odometry and/or vehicle speed. In particular, this Arduino is responsible for reading the calibrated data from the accelerometer, as well as the measurements from the Hall effect sensors in the rear tires of the vehicle.

The main cycle begins by checking if there is anything on the serial port. If there are data and they are the string “AP”, the counters for the number of sixteenths of a turn of both Hall effect sensors are reset.

Subsequently, the accelerations of the MPU6050 sensor were calculated, and they were calculated with an interval whether each of the Hall effect sensors in the tires has detected a neomid magnet or not.

As in the rest of the Arduino Uno devices, as a way of synchronizing with the master device, the library <TimerOne.h> was used to create a Timer1 object, and the attachInterrupt function was used to assign an interrupt every hundred milliseconds using a clock that guarantees that every hundred milliseconds are sent through the serial port to the master program: in this case, the accelerations and odometers of both tires.

### 3.4. Master Raspberry Pi 3

The Raspberry Pi 3 device, which hosts the master program implemented in Python3, begins with the declaration of libraries, necessary headers, and global variables.

Subsequently, the serial communication buffers are cleared for each of the Arduino Uno, and four threads are initialized using Thread objects from the threading library.

The first of the four threads executes the Update function, wherein the in_waiting function of the serial objects is utilized to determine if any data have been received from any Arduino Uno. If data are detected, they are formatted and used to update global variables.

The second thread executes the Write function, wherein a button press triggers the creation of a .csv file. Every 100 milliseconds, the current values of the following variables are logged into the file:Hour, minutes and seconds.Throttle position.Brake position.Steering wheel position.Magnetometer angle.Left rear odometer measurement.Right rear odometer measurement.Measurement of the accelerometer with respect to the X axis.Accelerometer measurement with respect to the Y axis.Total distance traveled by the vehicle.Current vehicle speed.Position of button one.Position of button two.Position of button three.Location of the vehicle on the X axis with respect to the inertial reference frame.Location of the vehicle on the Y axis with respect to the inertial reference frame.Initial orientation angle of the vehicle.Current test number.Auxiliary values related to the control algorithm being used.

The third thread executes the Show function, which alternately displays two sets of values every hundred milliseconds. The first set includes the values of speed, distance traveled, magnetometer angle, and orientation of the front tires. The second set includes the accelerometer values.

The fourth thread of the main program is responsible for determining the desired values for the accelerator, brake, and steering wheel and sending them to the Arduino Uno. This thread, known as the control thread, implements the control strategies, which will be further explored.

### 3.5. Cable Management

In any project, it is very useful to keep organized wiring, as this facilitates maintenance. If there is a problem or changes need to be made, having cables organized and clearly identified allows problems to be found and fixed more quickly and efficiently.

Organized wiring is not only functional but also improves the aesthetics of the environment. In addition, it reduces the risk of tripping, prevents possible damage from loose cables, and facilitates quick identification in an emergency [7].

All the devices mentioned in the previuos section were placed inside a metal cabinet at the rear of the vehicle. Additionally, an orange corrugated hose was used to hide the wiring underneath the vehicle.

An image of the metal cabinet can be seen in Figure 11.

It was decided to use Ethernet cables to connect each Arduino Uno with the sensors used and the H bridges. To achieve this, four phenolic boards were designed (one for each Arduino Uno), and a CNC was used to mark the necessary tracks and perforations, as can be seen in the Figure 12.

In total, ten Ethernet cables and ten connectors were designed for the sensors that connect directly to the Arduino Uno devices, which were organized as seen in Figure 13.

The connections and pins used on the devices are described below:HBAC: Connection between the H bridge and Arduino 1. Throttle control.
Arduino
PinColorH Bridged11OrangePWM Ld6Blue/WhiteEN Rd10BluePWM Rd7GreenEN LPOAC: Connection between the potentiometer and Arduino 1. Throttle control.
Arduino
PinColorPotentiometer5vGreenVCCGNDBlue/WhiteGNDA0Green/WhiteSignalHBFR: Connection between the H bridge and Arduino 2. Brake control.
Arduino
PinColorH Bridged11Green/WhitePWM Ld6Blue/WhiteEN Rd10BluePWM Rd7GreenEN LPOFR: Connection between the potentiometer and Arduino 2. Brake control.
Arduino
PinColorPotentiometer5vGreenVCCGNDBlueGNDA5Green/WhiteSignalHBVO: Connection between the H bridge and Arduino 3. Steering control.
Arduino
PinColorH Bridged11BluePWM Ld6Green/WhiteEN Rd10GreenPWM Rd7Blue/WhiteEN LBRVO: Connection between the magnetometer and Arduino 3. Steering control.
Arduino
PinColorMagnetometer5vBlue/WhiteVCCGNDGreenGNDi2C ClockGreen/WhiteSCLi2C DataBlueSDAENVO: Connection between the rotational encoder and Arduino 3. Steering control.
Arduino
PinColorRotational Encoderd5Green/WhiteDTGNDBlueGNDd4GreenCLKd2Blue/WhiteVCCOLVE: Connection between the Hall effect sensor of left tire and Arduino 4. Odometry.
Arduino
PinColorHall Effect SensorGNDGreenVCC5vBlueGNDA1OrangeSignalORVE: Connection between the Hall effect sensor of right tire and Arduino 4. Odometry.
Arduino
PinColorHall Effect SensorGNDGreenVCC5vBlueGNDA0Green/WhiteSignalACVE: Connection between the acelerometer and Arduino 4. Odometry.
Arduino
PinColorMagnetometer3vCafé/WhiteVCCGNDBlueGNDi2C ClockGreen/WhiteSCLi2C DataOrangeSDA

It is important to note that both the LCD screen and the electrical switches were connect directly to the Raspberry Pi 3 device and therefore did not require the development of a connector.

## 4. Modeling, Parametric Identification, and Simulations

In works related to autonomous vehicles [8,9] it is common to know the model of the vehicle that is purchased or adapted, with its parameters contained in its specifications. However, since it is an experimental vehicle, before thinking about control, there is both the problem of modeling it and determining the parameters that characterize said model.

While the mass of the vehicle after equipping can be determined using a floor scale, and some model parameters can be estimated by direct measurement, other critical parameters related to kinematic and dynamic models are not available. These parameters include the effective radius of the rear wheels, the ratio of the steering wheel angle to the angle of the front wheels, the longitudinal position of the center of mass, the yaw moment of inertia and the stiffness of the front and rear wheels, both longitudinal and lateral.

In this project, a dynamic model for a vehicle with Ackerman steering conditions was developed. Although the vehicle is physically configured with four wheels, the model adopted the simplified representation of a bicycle model consisting of a single concentrated wheel at the front and one at the rear. As the main contribution of this work, a series of straightforward tests were devised to facilitate the parametric identification of this dynamic model. The methodology used can be seen in Figure 14.

First, the a priori information about the vehicle geometry, the Ackermann condition, and some mechanical hypotheses were made, such as the relationship between the position of the steering wheel and the orientation of the tires or about the forces acting on the tires. The vehicle was then modeled from a kinematic and dynamic point of view.

Subsequently, a series of tests were designed to allow for the parametric identification of the dynamic model. These tests were carried out, and with the measurable variables of said tests and the known parameters, the parametric estimation was carried out. Lastly, a series of additional tests were used to validate the model and the parameters, thus ensuring that the result was an adequate model.

The development of the dynamic vehicle model, along with the conducted tests for parametric identification and experimental validation, has been submitted for review in another journal. In summary, a dynamic bicycle model was employed, thus incorporating specific considerations that have yielded the following equations:(1)$$\begin{array}{c} \dot{v_{x}}=v_{y}\dot{\psi }+\frac{2}{m}\left(F_{d}-F_{f}sin\left(\delta \right)\right)\\ \dot{v_{y}}=-v_{x}\dot{\psi }+\frac{2}{m}\left(F_{f}cos\left(\delta \right)+F_{r}\right)\\ \ddot{\psi }=\frac{2}{I}\left(l_{f}F_{f}cos\left(\delta \right)-l_{r}F_{r}\right) \end{array}$$
where
(2)$$\begin{array}{c} F_{d}=C_{\delta }\frac{v_{\omega }-v_{x}}{v_{\omega }}\\ F_{f}=C_{f}\left(\delta -\frac{v_{y}+l_{f}\dot{\psi }}{v_{x}}\right)\\ F_{r}=C_{r}\left(-\frac{v_{y}-l_{r}\dot{\psi }}{v_{x}}\right) \end{array}$$

A complete list of the variables, parameters, and symbols used in the vehicle model described by equations can be seen in Table 1.

The parameters that needed to be estimated in the vehicle model to be estimated were $r_{a}$, $k_{s}$, $C_{\delta }$, $C_{f}$, $C_{r}$, $l_{r}$, $l_{f}$, and *I*. Table 2 shows the parameters and the estimated value when performing the tests.

Once the mathematical model was validated with external tests and the error regarding the position predicted by it versus the actual measured position was small, it was possible to carry out simulations of the vehicle that allowed us to predict its behavior.

One of the main advantages of our proposal is that under said dynamic model and the proposed series of tests, it is possible to carry out the parametric identification without the need to use expensive equipment or perform sophisticated maneuvers. The described process only uses low-cost devices, and the proposed tests do not require complex control as in similar works like [10].

## 5. Trajectory Tracking

### 5.1. Test Trajectories

Before implementing the algorithms for tracking trajectories, it is necessary to precisely define which trajectories we are going to use in our tests. These trajectories provide an objective standard for measuring a controller’s ability to track trajectories.

The set of test trajectories was chosen based on the practical usefulness of these trajectories in autonomous driving; initially, the following of straight-line trajectories was also considered; however, due to its simplicity, in the end only the following trajectories were considered for comparison:Lane Change: While the average distance between one lane and another on a highway can vary depending on different factors, in general, the distance between opposing lanes is usually around 3 to 3.5 m. It was decided to use as a test trajectory the change from one lane to another located 3 m away carried out over a longitudinal distance of 4 m. Although the appropriate longitudinal distance to make a lane change depends on several factors, such as vehicle speed, traffic conditions, road regulations and general safety. Four meters was found to be an acceptable distance that simplifies operations, since it is produces a right triangle with a hypotenuse of 5 m. The lane change trajectory is shown in Figure 15.U-Turn: In general, the width necessary for a U-turn must allow the vehicle to perform the maneuver safely and completely, thus avoiding collisions with other vehicles or obstacles. In many places, traffic regulations specify that a U-turn must be made within a certain minimum radius; however, this information was not found in the specific case of Mexico. It was decided, considering the geometry of the vehicle and therefore its viability, to use a U-turn with an amplitude of 6 m. The U-turn trajectory is shown in Figure 16.Right Angle Curve: In general, a right-angle curve, also known as a 90-degree curve, is taken with as wide a radius as possible to allow for a smooth, safe turn. The radius of a 90-degree curve may vary depending on the road design, environmental restrictions, and local traffic regulations. It was considered to use a radius of 3 m such that the first part of the curve is equivalent to half of the U-turn considered in the previous example. Without loss of generality, only the left turn was considered, since the other case is analogous, as the vehicle and direction are symmetrical. The right angle curve trajectory is shown in Figure 17.

### 5.2. Control Algorithms

In this work, only geometric trajectory tracking techniques widely used in the literature, classical control techniques (PID), and a form of basic model predictive control (MPC) were considered.

Variants of nonlinear predictive control recently used in the literature such as in [11] were not considered, since, although they work very well in simulation, they are computationally expensive (with respect to both the Euler step and the minimization of the optimization problem that must be solved to find control input).

As it is a real physical prototype, a very fast real-time response is needed, which is why the use of techniques with low computational cost was preferred.

Intelligent control techniques like [12] were also not considered, because they tend to be more complex and, therefore, require greater processing power, which in this case is a limited resource. These kinds of techniques usually have variable execution times and outputs, which is not desirable in critical applications like this (consistent and deterministic outputs are preferred).

#### 5.2.1. Classic PID

The PID controller can be applied to trajectory tracking in autonomous vehicles to control the direction and keep the vehicle on a desired trajectory. There are many ways to calculate the error and implement a PID in a vehicle; for example, the current position of the vehicle can be compared with the target position on the trajectory or, as decided, the cross track error can be used as an error signal. (CTE). The CTE is the distance between the vehicle and the trajectory in general (not at a specific point).

The proportional component adjusts the vehicle’s steering based on the current error, the integral component corrects accumulated errors over time, and the derivative component improves the system’s response to rapid changes in error. By adjusting PID controller parameters such as the P, I, and D coefficients, smooth and accurate trajectory tracking can be achieved in an autonomous vehicle [13].

The main challenge lies in calculating the CTE (Crosstrack Error), which, due to limited processing time, necessitates computing the distance from the current position to each line segment formed by two consecutive points. It is impractical to employ a very fine discretization of the trajectory and calculate the distance from every point on that discretization to the current point, thereby owing to processing time constraints.

This is achieved by using the triple dot product to project the current position on each line segment generated by two consecutive points on the desired path and calculating their distance. When traversing all the points, the smallest distance will be the distance between the point and the path.

That is, for each pair of consecutive points $p_{1}\left(x_{1},y_{1}\right)$ and $p_{2}\left(x_{2},y_{2}\right)$ on the test path, it is required to do the following:Calculate the dot product between the vector with the current global position p(gx,gy) and the vector $v=p_{2}-p_{1}$.Calculate the length of the vector *v*.Calculate *t* as the dot product obtained in step 1 divided by the length of *v* (obtained in step 2) squared; this represents the perpendicular projection of *v* on *p*.According to the value of the parameter *t*, the distance between the global position *p* and the line segment between $p_{1}$ and $p_{2}$ are calculated.If the distance is the smallest of all those found so far, it is stored.

To determine the distance from the parameter *t*, three cases are considered:t<0: In this case, the closest point to *p* is $p_{1}$, and the distance between *p* and the line segment is the distance between *p* and $p_{1}$;t>1: In this case, the closest point to *p* is $p_{2}$, and the distance between *p* and the line segment is the distance between *p* and $p_{2}$;0<t<1. In this case, the closest point between *p* and the line segment is not at the ends but at the internal point $p_{1}+t\left(p_{2}-p_{1}\right)$.

Once this distance is calculated, the classic control algorithm performs the following steps:The error is defined as the smallest distance obtained with the previous algorithm.The distance traveled by the vehicle and its current position are used to choose a target point on the trajectory.The angle of the car with respect to said target point (obtained with the atan2 function) is used to determine the direction of rotation of the steering wheel.The error *e*, its derivative de, and its integral ie are calculated to calculate the magnitude of the steering wheel rotation as follows: $K_{p}e+K_{i}ie+K_{d}de$.The desired angular value from the previous step is transformed into a value in the steering wheel orientation interval (−38.38) and sent to the Arduino in charge of its control.

#### 5.2.2. Pure Pursuit

The pure pursuit controller algorithm has proven to be effective for trajectory tracking in autonomous vehicles, as it allows for a dynamic and precise response in real time [14].

This controller calculates a target point on the target trajectory that is a predetermined distance (called search distance) from the vehicle. This point is used to guide the vehicle onto the path. The steering angle required to reach the target point is determined using the curvature of the trajectory at the target point. In each iteration, the target point and steering angle are recalculated, thus allowing for accurate tracking of the target trajectory even in the presence of disturbances or changes in the environment.

In Figure 18, the red dot represents the target point. The distance between the rear axle and the target point is denoted as ld. Our objective is to orient the vehicle towards the correct angle and subsequently proceed towards this point. The angle formed between the vehicle body’s direction and the line of anticipation is termed α. Given that the vehicle is a rigid body and moves along a circular path, there exists an instantaneous center of rotation (ICR) with a radius of *R*.

Using the law of sines for the triangle in Figure 18
(3)$$\frac{ld}{sin(2\alpha )}=\frac{R}{sin(\pi /2-\alpha )}$$
it is possible to solve for *R* as
(4)$$\frac{ld}{2sin(\alpha )cos(\alpha )}=\frac{R}{cos(\alpha )}$$
(5)$$\frac{ld}{sin(\alpha )}=2R$$
so that using the definition of curvature *k*
(6)$$k=\frac{1}{R}=\frac{2sin(\alpha )}{ld}$$
and taking advantage of the fact that under the bicycle model from the previous section R=L/tan(δ), it can be deduced that the desired angle for the orientation of the front tires is
(7)δ=arctan(2Lsin(α)/ld).

In general, the pure pursuit geometric control algorithm performs the following steps:The distance traveled by the vehicle and its current position *p* are used to choose a target point on the trajectory po.The angle of the car with respect to ld is calculated as a=atan2(p−po).The desired angle for the front tires is calculated as δ=arctan(2Lsin(α)/ld).The desired angular value from the previous step is transformed into a value in the steering wheel orientation interval (−38.38) and sent to the Arduino in charge of its control.

The main challenge of this algorithm is determining the target point on the trajectory (and therefore the appropriate value of ld). It is reasonable to assume that the distance along the trajectory to which we are going to direct the vehicle depends on the speed.

Preliminary tests showed that indeed, if set for high speeds, the controller would converge to the trajectory very slowly at low speeds, while if set for low speeds, the controller would be dangerously aggressive at high speeds.

It was proposed, instead of a fixed distance ld, to use one that depends on the current speed of the vehicle, that is, $ld=k_{1}+k_{2}*V$ so that optimal performance can be achieved at both high and low speeds.

#### 5.2.3. Stanley Method

The Stanley driver is a geometric driver just like pure pursuit. It is the trajectory tracking approach used by the Stanford University team in the Darpa Grand Challenge [15]. Unlike the pure pursuit method that uses the rear axle as a reference point, the Stanley method uses the front axle as a reference point, as seen in Figure 19. At the same time, it takes into account both orientation error and Crosstrack Error (CTE). In this method, the CTE is defined as the distance between the closest point on the trajectory and the front axle of the vehicle, which is exactly the same as in the classic PID controller.

As can be seen in Figure 19, ψ(t) is the angle between the direction of the trajectory and the direction of the vehicle. The steering angle is represented as δ. The Stanley method has two intuitive laws of direction. First, the orientation error is eliminated by
(8)δ(t)=ψ(t).

Afterwards, the crosstrack error is eliminated; this is achieved using
(9)δ(t)=arctan(ke/v)
where e(t) is the crosstrack error, *v* is the speed, and *k* is a design constant.

The final step involves integrating both laws and introducing a slack term in the denominator, denoted as $k_{s}$, to prevent divisions by zero or nearly zero values, especially at low speeds. This adjustment yields the following:(10)$$\delta \left(t\right)=\psi \left(t\right)+arctan\left(\frac{ke}{v+k_{s}}\right).$$

Given that we already have knowledge of the Crosstrack Error (CTE) from its use in the classic PID controller, implementing the Stanley algorithm is not overly complex. The implemented control algorithm follows these steps:The heading error ψ is calculated.The CTE, denoted as *e*, is calculated using the same algorithm as in the classical control.The desired angle for the front tires is calculated as $\psi \left(t\right)+arctan\left(\frac{ke}{v+k_{s}}\right)$.The desired angular value from the previous step is transformed into a value in the steering wheel orientation interval (−38.38) and sent to the Arduino in charge of its control.

The main challenge of such an algorithm is to determine the appropriate values for $k_{s}<<1$ and $k_{e}$.

#### 5.2.4. Simple MPC

An MPC controller works as follows: First, a mathematical model of the vehicle is established that describes how the system evolves over time in response to control inputs. Then, a prediction horizon is defined that specifies the number of steps in the future that will be taken into account to make control decisions [16].

At each time step, the MPC controller uses the system model to predict how it will behave over the prediction horizon given the current control inputs and initial conditions. Next, a sequence of control actions is generated that is optimized to minimize a specified objective or performance criterion, such as following a desired trajectory, maintaining variables within certain ranges, or minimizing energy consumption.

However, instead of applying all the control actions in the sequence at once, the MPC only implements the first control action in the system and then repeats the process at the next time step. This allows the controller to continually readjust its control strategy as system measurements are updated and new information is obtained.

It was decided to use the bicycle model to predict the future moments of the vehicle to reduce the computational complexity and the system response time. In addition, only close values that could be reached on the steering wheel were considered in each future position.

Because the steering wheel positions are discrete and for each subsequent instant it can only reach a small number of positions, the optimization problem can be solved exhaustively.

The implementation of the implemented MPC algorithm performs the following steps:A prediction of *h* instants in the future is iteratively made considering only values of the steering wheel in the interval of its current position $\delta_{v}\pm 3$.For each set of values, the distance at which each step is from the desired position at that instant is calculated.A performance index is calculated as $j=\sum_{i=1}^{h}Qe_{i}^{2}+R\delta_{i}^{2}$.If the value of *j* is the smallest, the first desired control input $\delta_{i}$ is stored.The desired angular value from the previous step $\delta_{i}$ is transformed into a value in the steering wheel orientation interval (−38.38) and sent to the Arduino in charge of its control.

In the case of the MPC controller, it is important to determine the proposed performance index constants (in this case *Q* and *R*) that produce adequate behavior both in error reduction and in the use of the steering wheel. However, by confining the control inputs to a range closely aligned with the current value, the significance of *R*, which penalizes the control signal, diminishes slightly. Therefore, it was decided to use a value several orders of magnitude smaller than in *Q* to give priority to keeping the tracking error small.

The fundamental challenge is to determine the size of the prediction *h*, thus seeking to ensure that it allows us an adequate response time and that it is also computationally accessible.

### 5.3. Optimizing the Parameters of the Algorithms

According to what was observed in the previous subsections, each of the implemented algorithms has a set of design constants that must be optimized so that the performance of the algorithms is adequate.

In the PID algorithm, there are three constants to optimize: $K_{p},K_{i}$, and $K_{d}$.In the pure pursuit algorithm, there are two constants to optimize: $k_{1}$ and $k_{2}$.In the Stanley algorithm, there are two constants to optimize: $k_{s}$ and $k_{e}$.

The MPC controller was not considered in this optimization, since in its case, the higher *h* is, the better the performance must be, and its limitation will be provided by the hardware on which it is implemented.

Because an accurate model of the system was developed in the previous section, it was decided to optimize the constants of the control algorithms using simulations.

The process to perform controller gain optimization can be reduced to the following steps:Clearly define a cost or fitness function for our optimization process.Initialize the constants to be optimized for the chosen algorithm.Define an optimization algorithm that uses the cost function from step one to optimize the algorithm constants using system simulation.Perform as many iterations of the optimization algorithm until some stopping criterion is satisfied.

For our optimization, the integral of the squared error, or ISE, was used as a cost function.
(11)$$ISE=\int_{0}^{T}e{\left(t\right)}^{2}\;dt.$$
where e(t) represents the position error, that is, the distance at which each point of the simulation trajectory is with each point of the desired trajectory.

This criterion is used very frequently, mainly because it penalizes the squared error accumulated over time. This means that a controller that minimizes the ISE tends to keep the error close to zero for a long period of time. This is desirable in many control systems, since the aim is to maintain precise tracking of the reference over time. Furthermore, the ISE is a mathematically simple and easy-to-calculate metric, thus making it convenient for controller analysis and tuning.

The algorithm used to carry out the optimization was the MATLAB function fminsearch, which is an implementation of the Nelder–Mead algorithm that was described in the theoretical framework.

The stopping criterion of the optimization algorithm was to consider a maximum of iterations equal to 200 for the number of constants to be optimized or that the cost function is not reduced by at least $1\times 10^{-4}$; these criteria are those recommended by the default of the fminsearch function, and in this case, they allow us to reach at least local optima.

The initial parameters of the constants were found experimentally, and multiple initial points were used with the help of the MATLAB rand function. After the optimization process, the best values can be seen in the Table 3.

## 6. Experimental Results of Control Algorithms

The results presented in this section are based on experimental data obtained from the physical prototype rather than simulations. Each trajectory depicted herein represents real-world observations, thus serving to validate the experimental performance of the prototype, not simulated scenarios. These experimental results offer insights into the practical behavior and capabilities of the developed vehicle model under real-world conditions.

### 6.1. Performance Criteria

Each of the control algorithms in each of the test trajectories was compared based on two performance parameters or criteria:The integral of the quadratic error. This criterion was defined in the previous section; in particular, the error to be integrated is the tracking error that represents the distance at which each point on the actual trajectory is from the desired trajectory.The norm of the vector representing the changes in the position of the steering wheel is determined as follows: first, the $\delta_{v}$ values are utilized to compute the vector $\Delta \delta_{v}$, and subsequently, the norm of this vector is evaluated.

This second criterion is used to evaluate how large and abrupt the steering turns are so that the control algorithms can also be compared in terms of the type of driving. Significant increases in steering wheel orientation $\delta_{v}$ equate to sudden movements that may be uncomfortable for passengers and may affect safety.

### 6.2. Lane Change

The average values of the ISE and $\left|\Delta \delta_{v}\right|$ after ten executions of each of the control algorithms on the desired lane change trajectory described in the previous section are shown in Table 4.

In Figure 20, Figure 21, Figure 22 and Figure 23, you can see an example of tracking the lane change trajectory using each of the control algorithms. The axes of the graphs represent the X and Y coordinates within an inertial reference frame external to the vehicle. Thus, the graphs illustrate the vehicle’s displacement in the XY plane, thereby accounting solely for movement within this plane.

### 6.3. Right Angle Curve

The average values of the ISE and $\left|\Delta \delta_{v}\right|$ after ten executions of each of the control algorithms on the right angle trajectories that were described in the previous section are shown in Table 5.

In Figure 24, Figure 25, Figure 26 and Figure 27, the right-angle turning trajectories are depicted in four different tracking examples. Figure 24 displays the results of employing classical control. Figure 25 illustrates the effects of using pure pursuit control. Figure 26 demonstrates the outcomes of utilizing Stanley control. Finally, Figure 27 illustrates the effects of employing Model Predictive Control.

### 6.4. U-Turn

The average values of the ISE and $\left|\Delta \delta_{v}\right|$ after ten executions of each of the control algorithms on U-turn trajectories that were described in the previous section are shown in Table 6.

In Figure 28, Figure 29, Figure 30 and Figure 31, you can see an example of tracking the U-turn trajectory using each one of the control algorithms.

## 7. Conclusions and Future Work

The purpose of this study was the development of an experimental prototype of an electric vehicle with a control system for trajectory tracking. The vehicle was successfully equipped and instrumented using only low-cost electronic devices, thus providing an economical alternative for the effective estimation of the positions, orientations, and speeds of an experimental vehicle.

The developed model has managed to effectively capture the dynamics of the vehicle and has allowed us to carry out simulations of the vehicle’s behavior in a satisfactory manner. A classic controller, MPC controller, pure pursuit, and Stanley geometric controllers have been implemented on the vehicle. The parameters of these controllers have been successfully optimized initially in simulation and subsequently on the experimental vehicle.

Finally, the experimental performance of the controllers in following three test trajectories—lane change, right angle curve, and U-turn—has been compared, according to the results of the previous section, in terms of both the ISE of the error of tracking, as well as the norm of the change in the steering wheel orientation; the MPC controller produced better results than the classic PID controller in all trajectories.

This performance is a result of the inability of the classical controller to predict errors in the future unlike the MPC controller. In classical control, control actions occur only after an error in orientation occurs and do not anticipate future errors like MPC-based control techniques.

The geometric controllers showed consistently better results than the classical control, because they use future positions on the trajectory to direct their direction in their formulation; however, it was difficult to find constants that produced good performance in all test trajectories and at different speeds. For example, the Stanley controller with the parameters found produces even better results in lane changing than the MPC controller; however, it was not as good in the case of U-turns.

It is also important to note that the MPC controller used a very short prediction horizon due to hardware limitations, since its implementation is very computationally expensive, unlike geometric controllers, whose simplicity and ability to use future positions in the desired trajectory make them a more than reasonable alternative for environments with limited computational resources.

With respect to the implementation of a longer prediction horizon, a more sophisticated version of MPC like the Fuzzy MPC [17], Deep Learning-Based NMPC [18], or artificial intelligence-based trajectory tracking algorithms like the Neural Network Trajectory Tracking [19] remains as future work, as it requires the use of a dedicated device, microcomputers, or additional hardware solutions.

This work is part of a larger project that aims to achieve self-driving, so it could also be considered future work to unite this scheme with a strategy for generating trajectories in real time. 

## Figures and Tables

**Figure 1 sensors-24-02769-f001:**
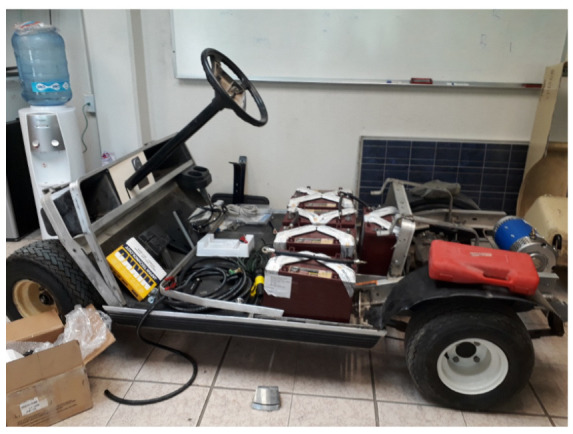
Chassis of the experimental vehicle.

**Figure 2 sensors-24-02769-f002:**
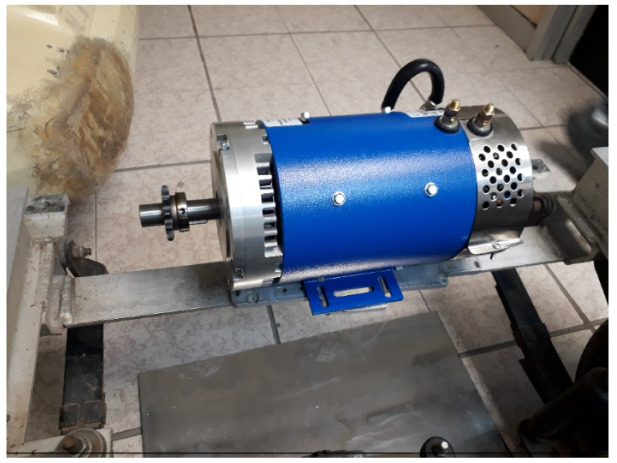
Prototype vehicle direct current motor.

**Figure 3 sensors-24-02769-f003:**
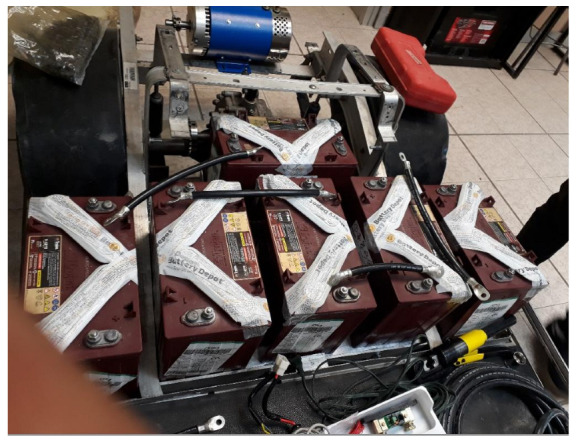
The six TROJAN batteries used to power the electric motor.

**Figure 4 sensors-24-02769-f004:**
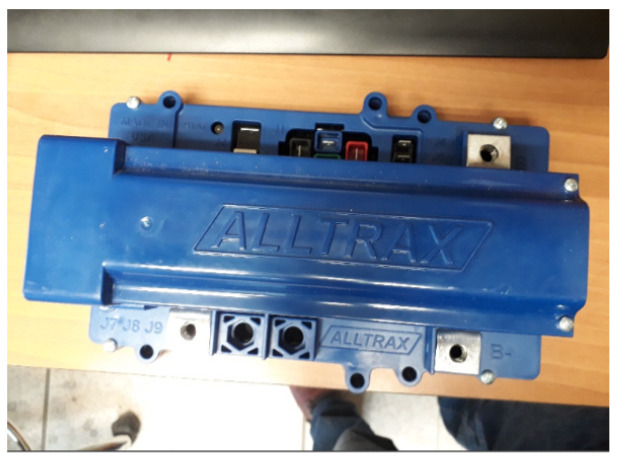
Voltage controller and control box.

**Figure 5 sensors-24-02769-f005:**
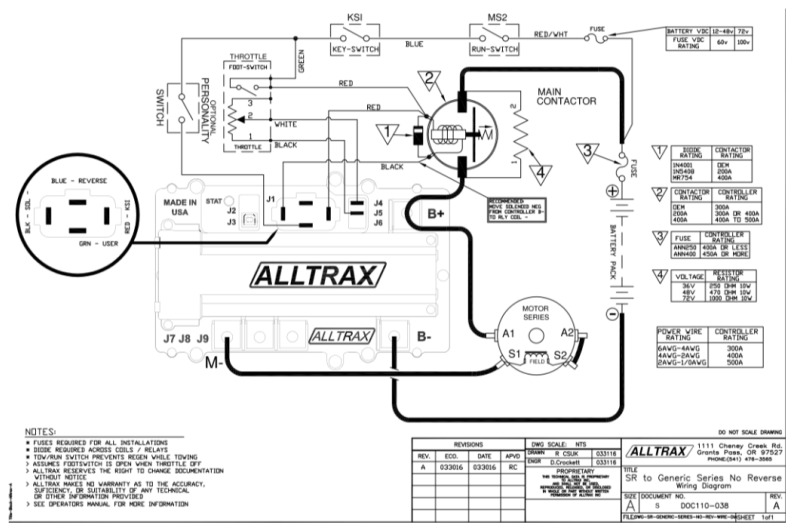
Connection diagram extracted from Alltrax Inc. See https://alltraxinc.com (accessed 2 April 2024).

**Figure 6 sensors-24-02769-f006:**
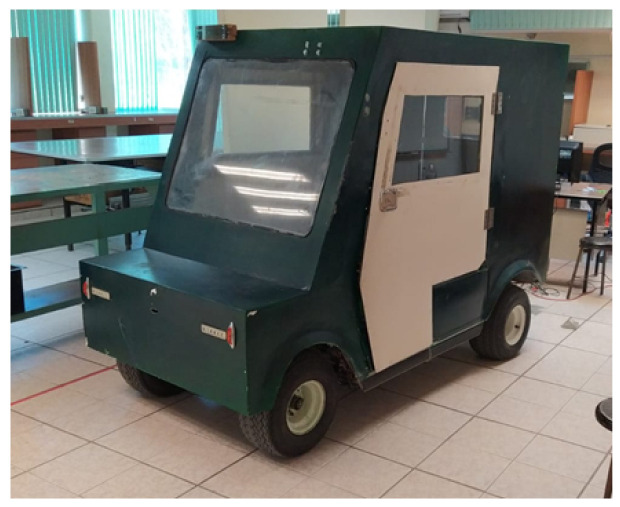
Prototype vehicle with bodywork.

**Figure 7 sensors-24-02769-f007:**
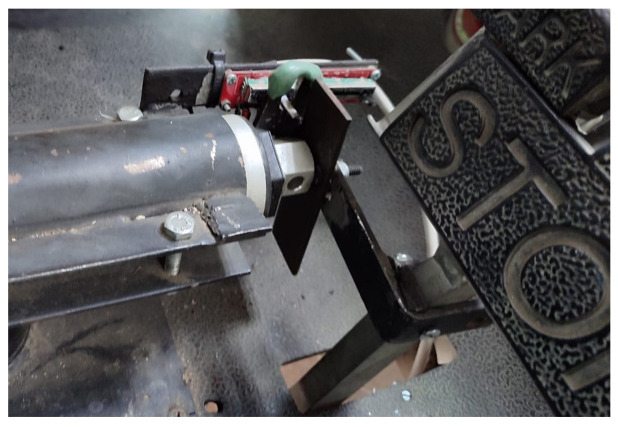
Linear potentiometer placed on the brake pedal.

**Figure 8 sensors-24-02769-f008:**
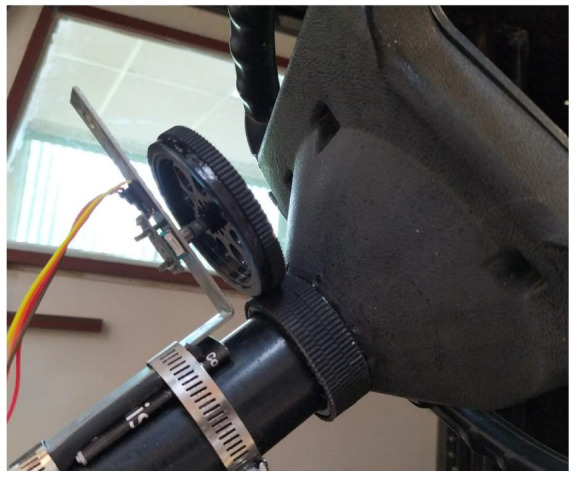
Rotational encoder placed on the steering wheel to measure angular displacement.

**Figure 9 sensors-24-02769-f009:**
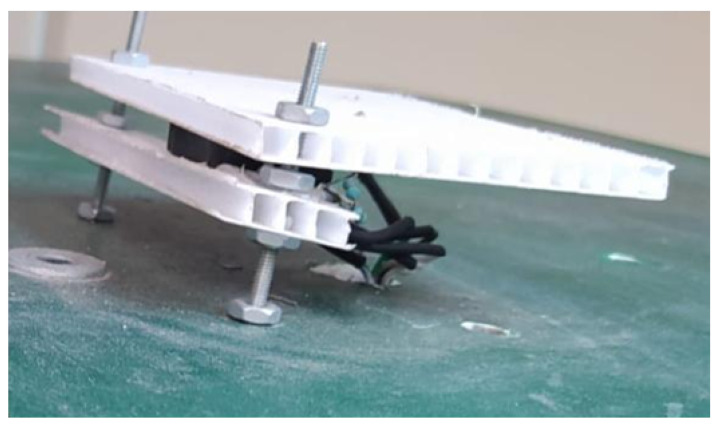
Platform with the magnetometer and accelerometer on top of the vehicle.

**Figure 10 sensors-24-02769-f010:**
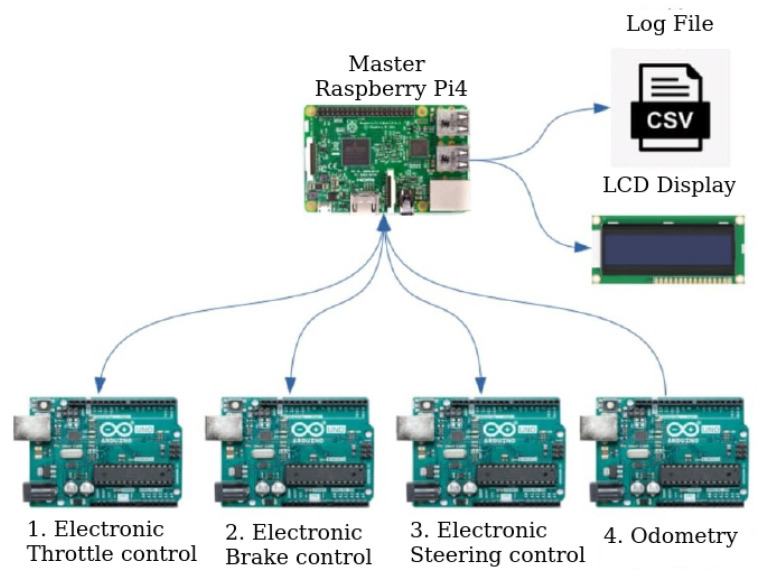
Communication scheme.

**Figure 11 sensors-24-02769-f011:**
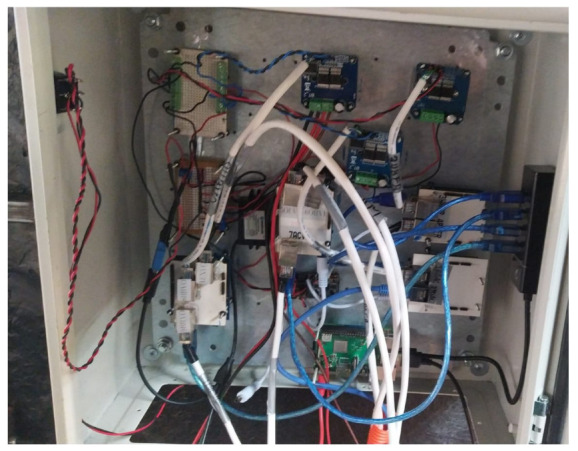
Metal cabinet at the rear of the vehicle.

**Figure 12 sensors-24-02769-f012:**
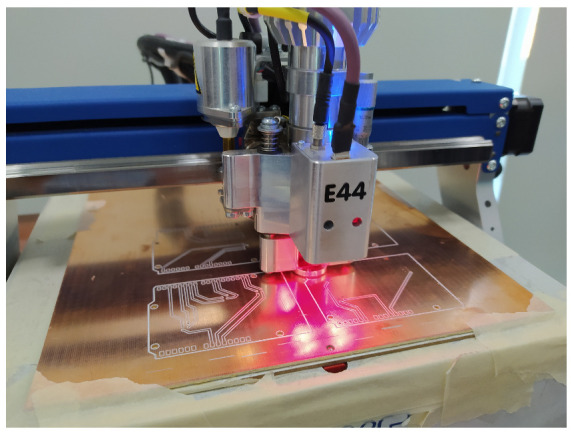
A CNC building the phenolic boards of the Arduino Uno.

**Figure 13 sensors-24-02769-f013:**
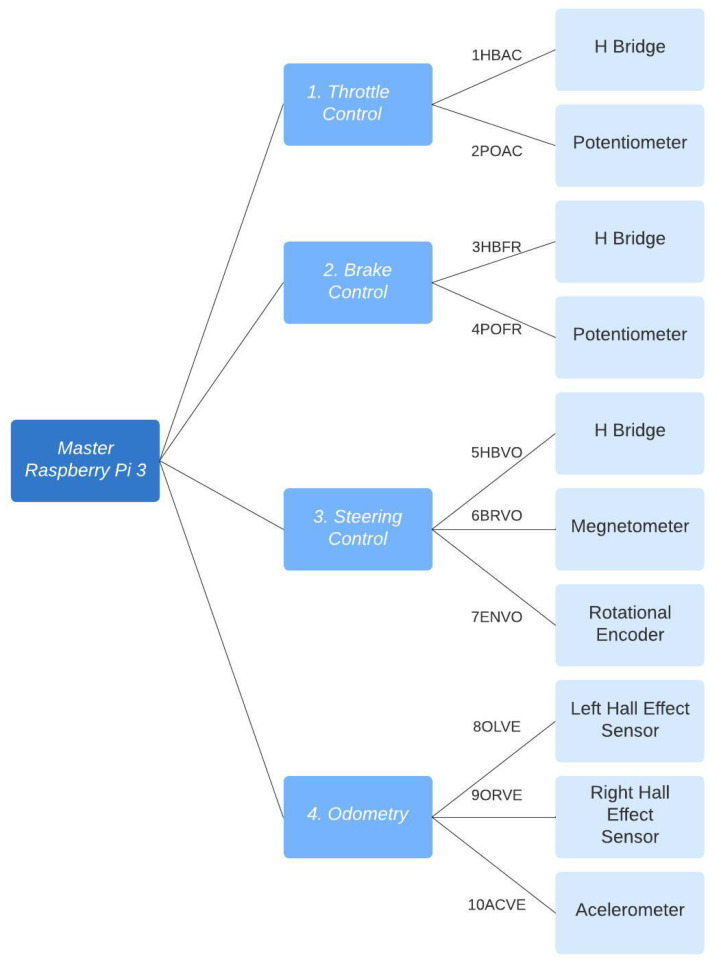
Diagram with the hardware devices and the ten connections.

**Figure 14 sensors-24-02769-f014:**
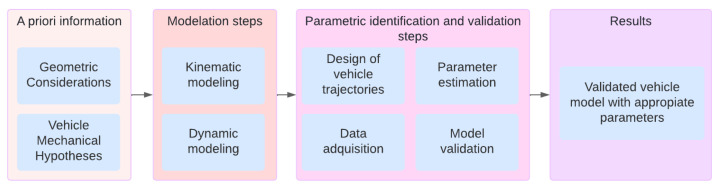
Proposal methodology to model the vehicle.

**Figure 15 sensors-24-02769-f015:**
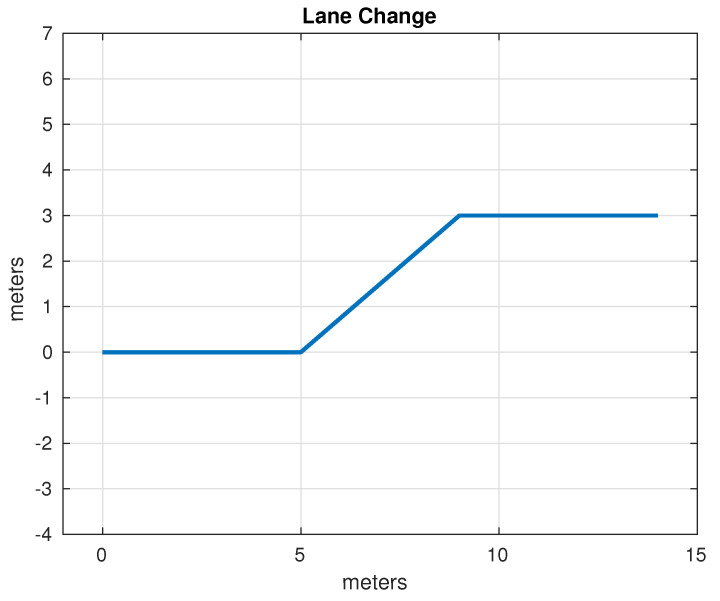
Test path: lane change.

**Figure 16 sensors-24-02769-f016:**
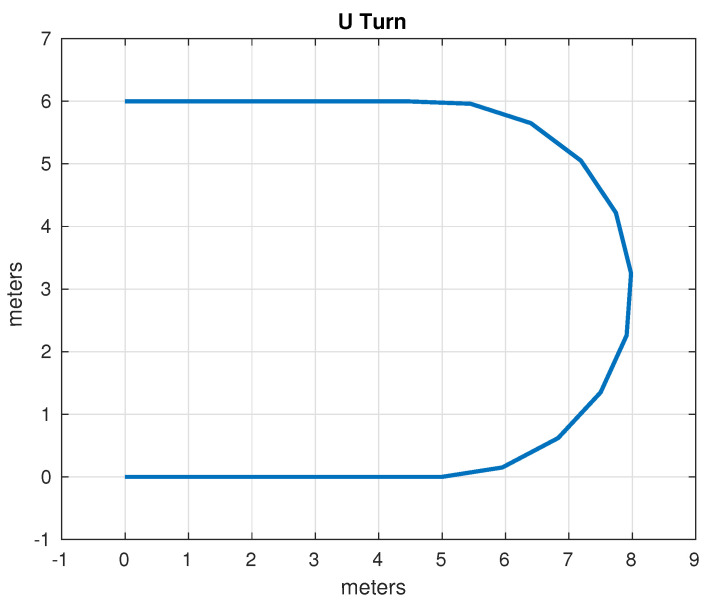
Test path: U-turn.

**Figure 17 sensors-24-02769-f017:**
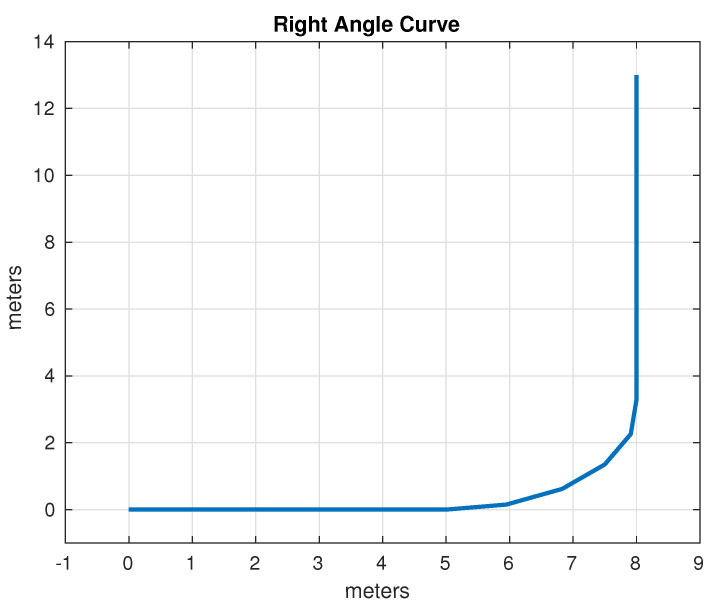
Test path: right angle curve.

**Figure 18 sensors-24-02769-f018:**
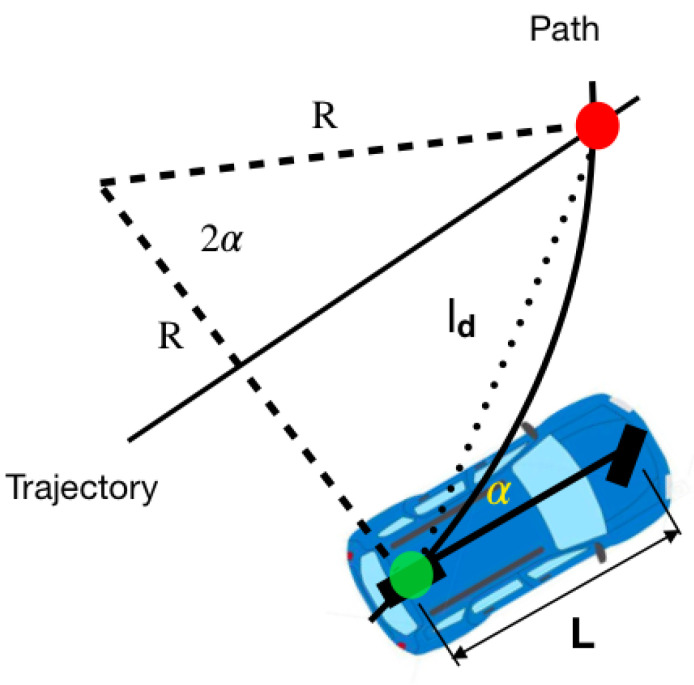
Geometry of the pure pursuit algorithm taken from Yan Ding Shuffle AI: https://www.shuffleai.blog/blog Accessed 2 April 2024.

**Figure 19 sensors-24-02769-f019:**
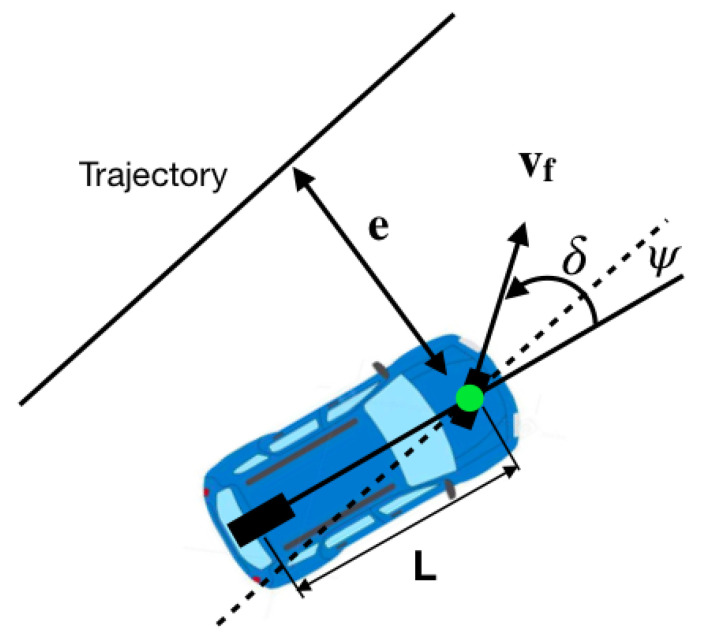
Stanley method geometry taken from Yan Ding Shuffle AI: https://www.shuffleai.blog/blog Accessed 2 April 2024.

**Figure 20 sensors-24-02769-f020:**
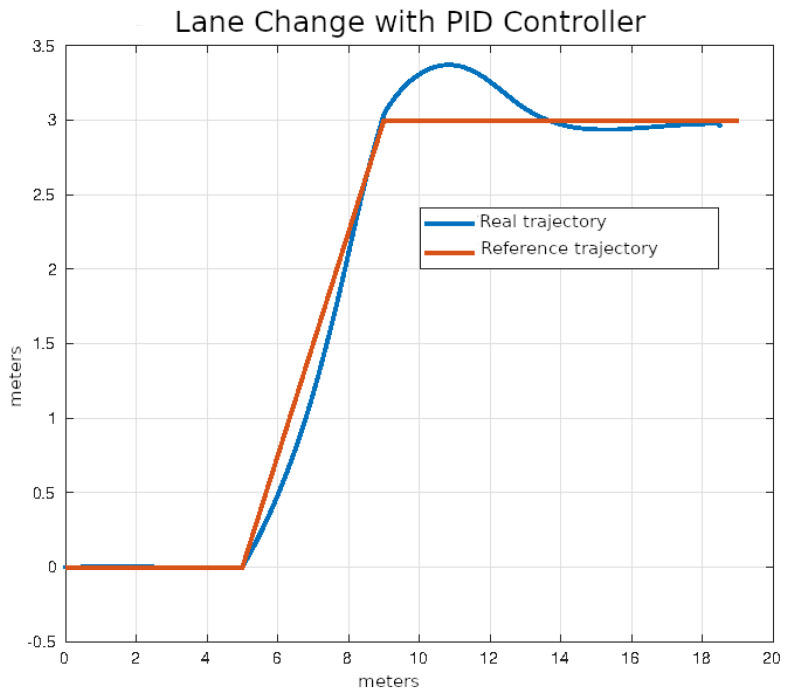
Lane change using classic control.

**Figure 21 sensors-24-02769-f021:**
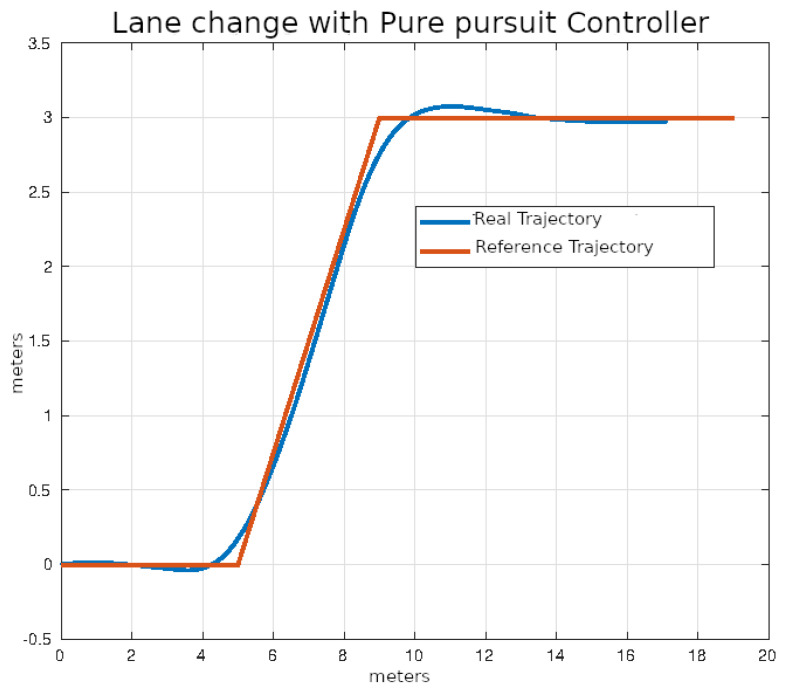
Lane change using pure pursuit control.

**Figure 22 sensors-24-02769-f022:**
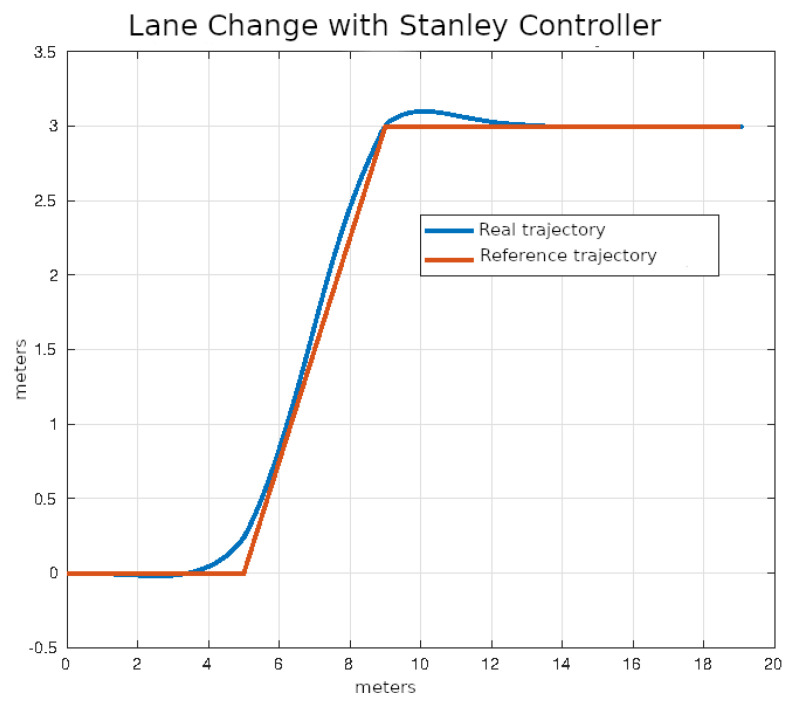
Lane change using Stanley control.

**Figure 23 sensors-24-02769-f023:**
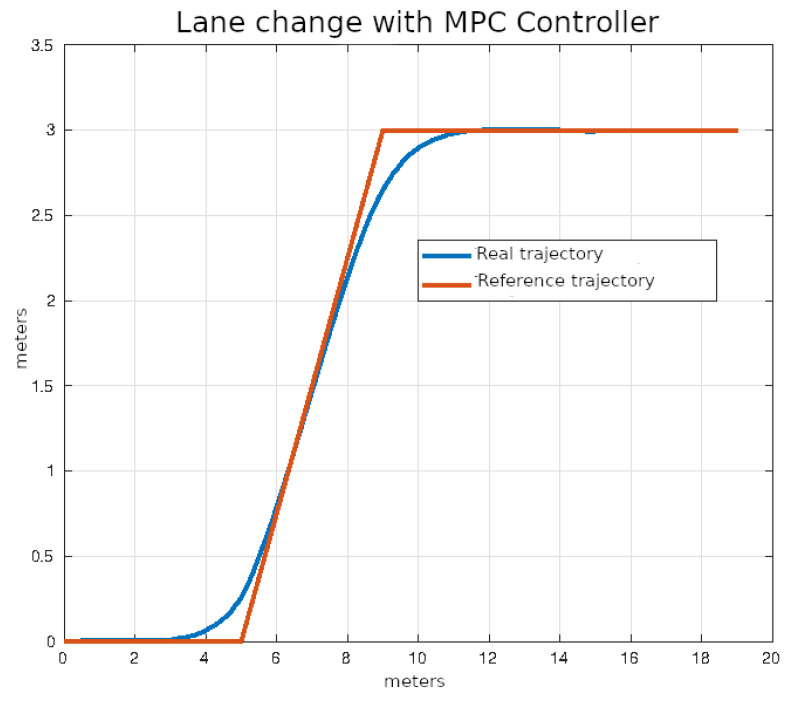
Lane change using MPC control.

**Figure 24 sensors-24-02769-f024:**
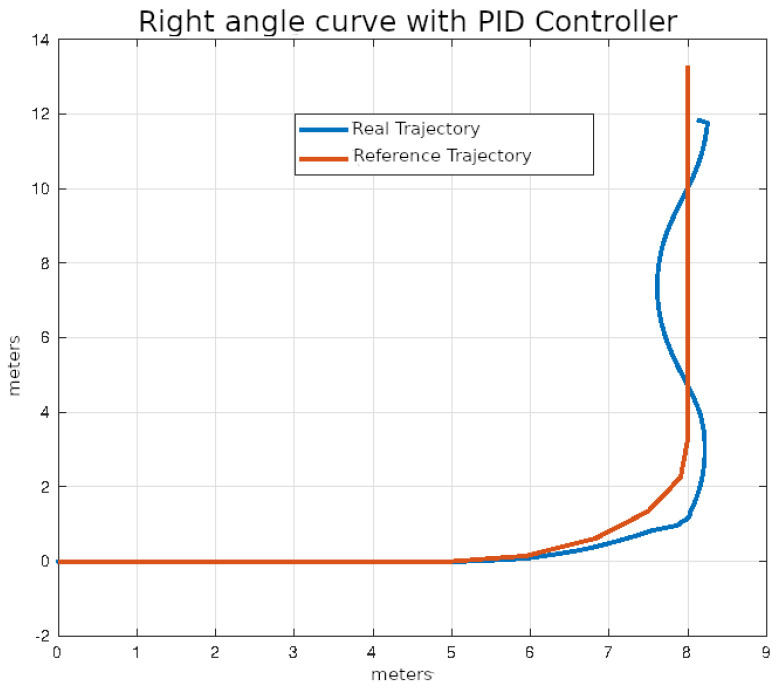
Right angle curve using classical control.

**Figure 25 sensors-24-02769-f025:**
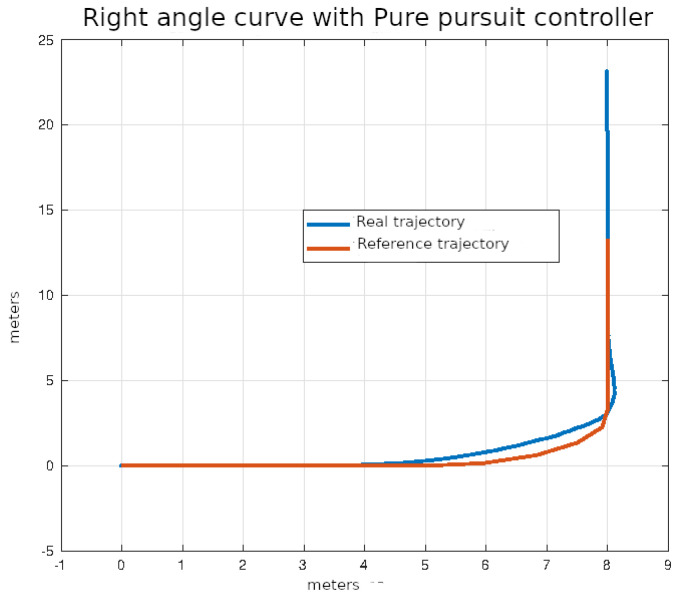
Right angle curve using pure pursuit control.

**Figure 26 sensors-24-02769-f026:**
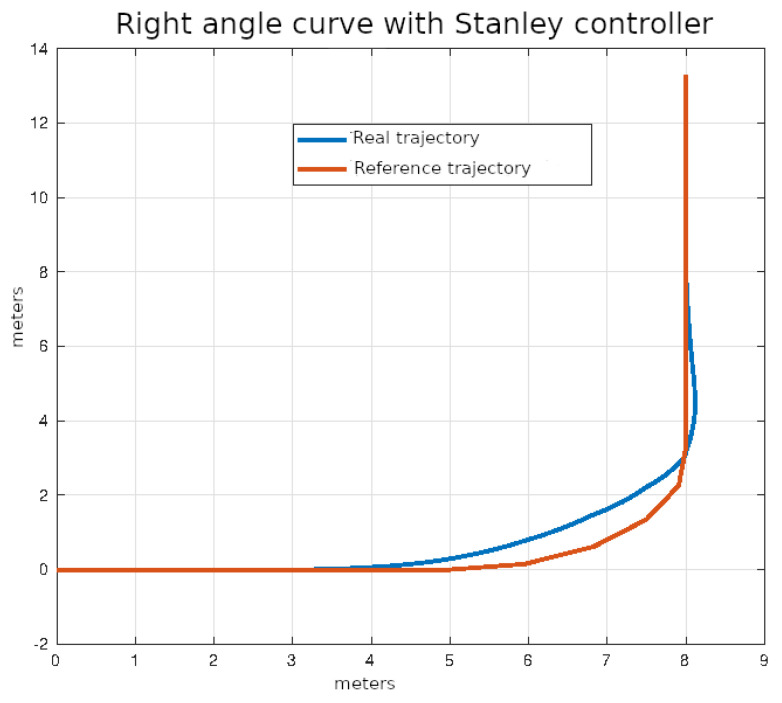
Right angle curve using Stanley control.

**Figure 27 sensors-24-02769-f027:**
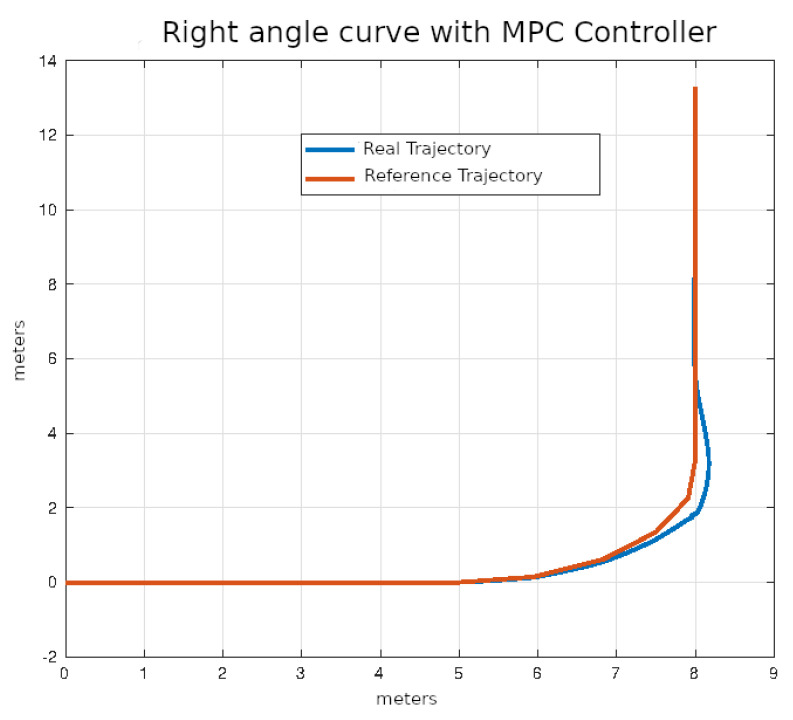
Right angle curve using MPC control.

**Figure 28 sensors-24-02769-f028:**
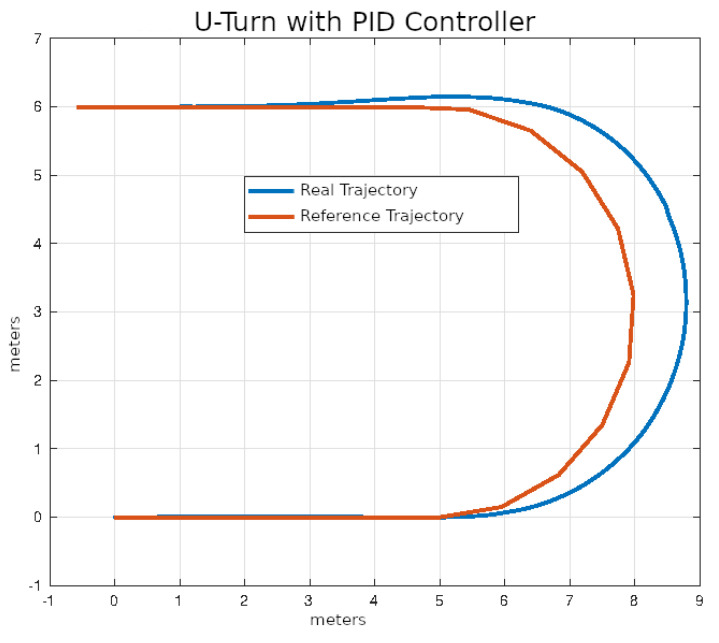
U-turn using classical control.

**Figure 29 sensors-24-02769-f029:**
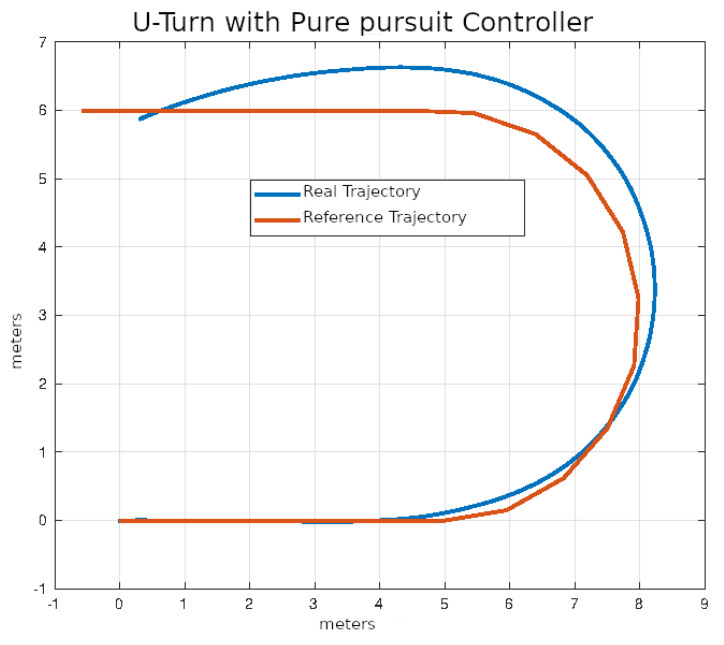
U-turn using pure pursuit control.

**Figure 30 sensors-24-02769-f030:**
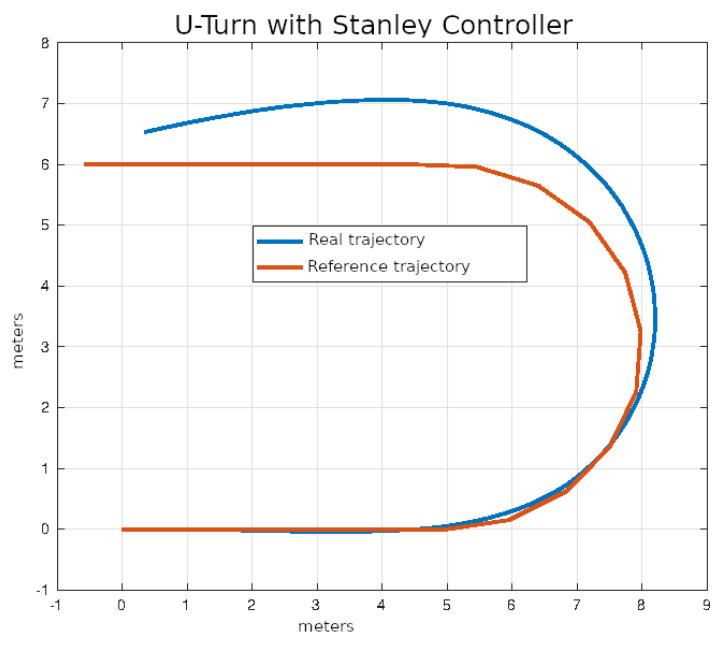
U-turn using Stanley control.

**Figure 31 sensors-24-02769-f031:**
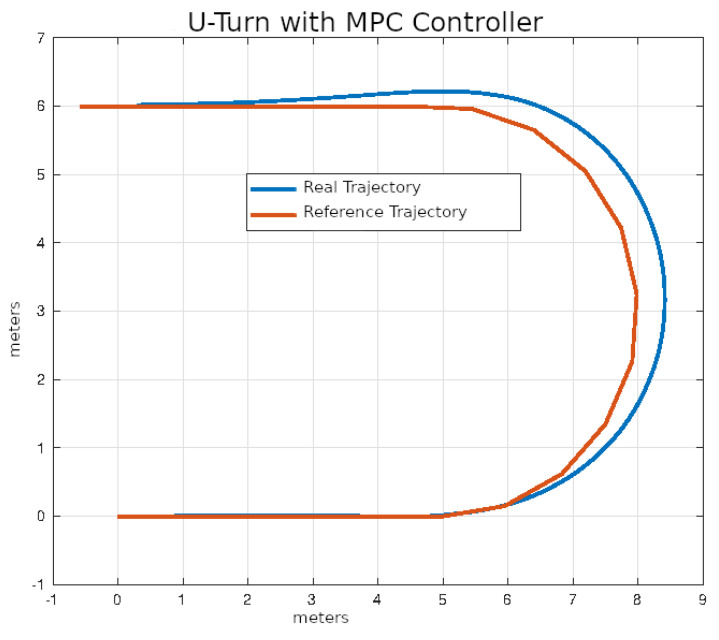
U-turn using MPC control.

**Table 1 sensors-24-02769-t001:** List of symbols.

Symbol	Description
$r_{a}$	Rear wheel’s actual radius.
$k_{s}$	Proportionality constant between δ and $\delta_{s}$.
$C_{\delta }$	Longitudinal tire stiffness of the rear wheels.
$C_{f}$	Lateral tire stiffness of the front wheels.
$C_{r}$	Lateral tire stiffness of the rear wheels.
$l_{r}$	Length from the left of mass to the rear axle.
$l_{f}$	Length from the left of mass to the front axle.
$B_{r}$	Length of the rear axle.
*I*	Moment of inertia with respect to the z axis.
*m*	Mass of the vehicle.
ψ	Angle of orientation of the vehicle.
$v_{\omega }$	Speed of the rear tires of the vehicle.
$v_{x}$	Speed of the vehicle body in the direction of the x axis.
$v_{y}$	Speed of the vehicle body in the direction of the y axis.
$\omega_{a}$	Average angular velocity of the vehicle’s rear tires.
δ	Steering angle of the front wheels.
$\delta_{s}$	Angle of the steering wheel.
$F_{d}$	Longitudinal force of the rear wheels (driving force).
$F_{f}$	Lateral force of the front wheels.
$F_{r}$	Lateral force of the rear wheels.

**Table 2 sensors-24-02769-t002:** Identified vehicle parameters.

Symbol	Value
$r_{a}$	0.211 m
$k_{s}$	0.1791
$C_{\delta }$	34.7596
$C_{f}$	67.4391
$C_{r}$	51.0454
$l_{r}$	0.8688 m
$l_{f}$	0.8311 m
*I*	488.43 kg m^2^

**Table 3 sensors-24-02769-t003:** Control algorithm parameters.

Strategy	Optimized Parameters
PID	$k_{p}=47.34$, $k_{i}=0.012$, $k_{d}=0.73$
Pure pursuit	$k_{1}=3.17$, $k_{2}=0.298$
Stanley	$k_{s}=0.00001$, $k_{e}=0.7$

**Table 4 sensors-24-02769-t004:** Performance of control algorithms when changing lanes.

Strategy	ISE	$\left|\Delta \delta_{v}\right|$
PID	0.5198	21.44
Pure pursuit	0.0917	16.85
Stanley	0.1294	17.69
MPC	0.1531	19.28

**Table 5 sensors-24-02769-t005:** Performance of control algorithms on right angle curves.

Strategy	ISE	$\left|\Delta \delta_{v}\right|$
PID	1.2873	19.84
Pure pursuit	0.5289	15.39
Stanley	0.4952	14.56
MPC	0.2895	14.76

**Table 6 sensors-24-02769-t006:** Performance of control algorithms in U-turn.

Strategy	ISE	$\left|\Delta \delta_{v}\right|$
PID	2.1456	16.28
Pure pursuit	1.2912	16.30
Stanley	1.5473	15.74
MPC	0.6324	14.45

## Data Availability

Data are contained within the article.

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
