# Peer review of "Design, Construction, and Validation of an Experimental Electric Vehicle with Trajectory Tracking"

_sensors, 2024, doi:10.3390/s24092769_

Round 1

Reviewer 1 Report

Comments and Suggestions for Authors

Comments on the Quality of English Language

no

Author Response

Dear Reviewer, 

Attached to this message you will find a word file with the detailed response to the comments you kindly sent us.

Reviewer 2 Report

Comments and Suggestions for Authors

This research presents an experimental electric vehicle developed at the Tecnológico Nacional de México campus Celaya. However, some following problems need to be clarified:

1. This paper mainly develops an experimental platform, please analyze the vehicle model built in this paper, compared with the existing models, what are the advantages?

2. How representative are the selected control algorithms? Please explain the advantages of optimal control over traditional nonlinear control methods. You can refer to the following papers for analysis:

https://doi.org/10.3390/electronics11071110

https://doi.org/10.1016/j.asoc.2020.106304

https://doi.org/10.1016/j.asoc.2020.106304

3. This paper employs a dynamic bicycle model, but it lacks the necessary controller design process when using the control algorithm to track the trajectory. Please supplement the designed controller and select parameters.

4. Please clearly explain the meaning of the horizontal and vertical coordinates of the simulation diagram in Figure 15-31.

5. The control curves of several algorithms can be placed on one graph, so that the simulation comparison is more intuitive.

Comments on the Quality of English Language

It is best to unify the names of the experimental vehicle built in the article.

Author Response

(The authors gave the same response as above.)

Reviewer 3 Report

Comments and Suggestions for Authors

The paper introduces the design of an electrical vehicle. This is an interesting topic to reveal the whole process of building a real-world machine. However, as a scientific paper, there are contents need to be revised. My suggestion is accept the paper after a minor revision. Here are my suggestions. 

1. The scientific contributions should be clearly stated in the introduction, especially why the work in this paper is more valuable than the others. Moreover, pls introduce the motivation of this research. 

2. The abstract could be improved by expanding the introduction of the methodology. 

3. The figures could be colored. 

4.  The literature review needs improving, here are some recommendations. 

10.1109/TVT.2019.2948153

10.1109/TITS.2021.3094738

10.1109/JPROC.2021.3072788

https://doi.org/10.3390/jmse12010126

10.1109/TVT.2022.3141732

Comments on the Quality of English Language

moderate

Author Response

(The authors gave the same response as above.)

Round 2

Reviewer 2 Report

Comments and Suggestions for Authors

We gratefully appreciate your positive comments and constructive suggestions.